# Satellite-Observed Time and Length Scales of Global Sea Surface Salinity Variability: A Comparison of Three Satellite Missions

Daling Li Yi [1,2], Oleg Melnichenko [2,3], Peter Hacker [2,4] and Ke Fan [1,*]

[1] School of Atmospheric Sciences, Sun Yat-Sen University, and Southern Marine Science and Engineering Guangdong Laboratory (Zhuhai), Zhuhai 519082, China
[2] International Pacific Research Center, School of Ocean and Earth Science and Technology, University of Hawai'i, Honolulu, HI 96822, USA
[3] Earth and Space Research, Seattle, WA 98105, USA
[4] Hawaii Institute of Geophysics and Planetology, School of Ocean and Earth Science and Technology, University of Hawai'i, Honolulu, HI 96822, USA
[*] Correspondence: fank@mail.sysu.edu.cn

**Abstract:** Sea surface salinity (SSS) observations from Aquarius, Soil Moisture and Ocean Salinity (SMOS), and Soil Moisture Active Passive (SMAP) satellite missions are compared to characterize the time and length scales of SSS variability globally. Overall, there is general agreement between the global patterns of the time and length scales of SSS variability estimated from the three satellite missions. The temporal scales of SSS variability vary from more than 90 days in the tropics to ~15 days in the Southern Ocean. The very short temporal scales (close to the Nyquist period) in some parts of the ocean are probably due to the high level of noise in the satellite data or the high noise-to-signal ratio. The longest temporal scales are observed along the South Pacific Convergence Zone (SPCZ) and in the central and western tropical Pacific. These areas are also related to the strongest ENSO-related signal in SSS. The processes governing the SSS variability and distribution are also non-stationary, such that the scales determined over different observation periods may differ. Dominant spatial scales of SSS variability are generally the longest (up to 150 km) in the tropics and the shortest (<60 km) in the subpolar regions. The distribution of the dominant spatial scales is not simply latitudinal but exhibits a more complex spatial pattern. In the tropics, there is slight east-west and inter-hemispheric asymmetry observed in the Pacific but absent in the other two oceans. The analysis also reveals that the length scales of SSS variability are highly anisotropic in the tropics (the zonal scales are generally shorter than the meridional ones) and become more isotropic towards higher latitudes. Regional differences in the estimates of the scales from the three satellite SSS datasets may arise due to differences in the observation duration, spatial resolution and/or different level of noise.

**Keywords:** sea surface salinity; decorrelation scale; Aquarius; SMAP; SMOS

## 1. Introduction

Sea surface salinity (SSS) is a key essential climate variable and reflects the intensity of the marine hydrological cycle [1,2]. Variations in SSS affect seawater density, stratification and mixing, significantly influencing the stability of the upper ocean [3]. In the tropics, SSS variability may affect the air-sea exchange of heat and momentum by means of a barrier-layer formation [4,5]. At high latitudes, sea water density is governed by salinity and the variations in SSS can control the occurrence and depth of deep convection [6]. Through its effect on density, salinity influences ocean circulation, which, in turn, is a key regulator of Earth's climate. Therefore, monitoring, quantifying, and characterizing SSS variability is essential for understanding the global water cycle, ocean circulation, and for climate monitoring and prediction [7].

SSS variability spans a broad range of time and length scales, from decadal (>10 years) to intraseasonal (<90 days), and from basin scales (>1000 km) to submesoscales (<10 km). One of the parameters characterizing SSS variability is the dominant scale of variability, which can be a temporal scale, characterizing variability in time, or a spatial scale, characterizing variability in space. The temporal scale is used to characterize and quantify the persistence of SSS anomalies, known as "ocean memory" and is often measured by autocovariance [8,9], while the spatial scale is a reflector of the physical processes behind the observed features [10–12]. They are also the scales at which the dominant patterns emerge, i.e., the scales with the climate leading mode and/or maximum spatial variance. In practical applications, knowledge of the time and length scales of SSS variability is necessary for the objective interpolation (OI) of irregularly spaced observations [13–15]. This is particularly important as this method is widely used to map satellite observations of SSS and blend observations from multiple satellite platforms (e.g., [16,17]).

For those reasons, assessment of the spatiotemporal scales of SSS variability has continued using various platforms and approaches, most of which are based on in situ observations [18]. Until recently, knowledge of SSS variability was limited by lack of observations and was obtained from a collection of sparse and irregular observations mainly monitored by ships of opportunity and research vessels [19–21]. Delcroix et al. [22] utilized ship-track observations collected over the period 1970–2003 to assess the time and space scales of SSS variability in the tropics. The spatiotemporal scales of SSS variability were found to vary geographically, reflecting the dominant physical processes controlling the variability. Because of the sparse data coverage, the study was limited to the tropics and a few repeat ship tracks. With the growth of the Argo program over the past two decades, the situation has improved significantly. The Argo array, however, provides one profile every 10 days over a $3° \times 3°$ box [23]. Those profiles inadequately resolve important SSS variability, including mesoscale and intra-seasonal variability, which may account for a great proportion of SSS variance. Using Argo, mooring, and other in situ salinity observations, Martins et al. [24] estimated spatiotemporal scales of SSS variability in the Atlantic Ocean. They found that the scales vary geographically between 100–250 km and between 30–70 days when the seasonal cycle is included. The study also pointed out the importance of short-scale salinity features and fast variability, not resolved by conventional Argo measurements.

The launch of three satellite missions, Aquarius, Soil Moisture and Ocean Salinity (SMOS) and Soil Moisture Active Passive (SMAP), have initiated a new era of studying salinity variability [25,26]. With their frequent revisit time and global coverage, satellite missions have provided an unprecedented opportunity to systematically investigate SSS variability from a global perspective. The first near-global estimate of the temporal and spatial scales of SSS variability from early Aquarius data (3+ years) was provided by Bingham and Lee [27]. They used gridded SSS fields to produce the estimates and found that in over half of the global ocean the decorrelation time scales were shorter than 80 days when the seasonal cycle was not removed. Removing the seasonal cycle resulted in even shorter scales (<80 days), dominating the global ocean. Bingham and Lee [27] found the spatial scales to be very short except in the tropical oceans influenced by the Intertropical Convergence Zone (ITCZ) and the South Pacific Convergence Zone (SPCZ). There, the spatial scales were found to be anisotropic with the zonal scales being slightly larger than the meridional scales, probably modulated by the horizontal advection [27]. To our knowledge, this is the only study that provides a global characterization of the dominant scales of SSS variability, yet it is based on a relatively short period of Aquarius observations (3+ years).

The accumulation of satellite SSS data over the past decade provides an opportunity to revisit the subject with more accurate estimates based on much longer time series of SSS measured by the three satellite missions. Here we present results from the analysis of 5 years of Aquarius satellite SSS measurements, 6 years of SMAP satellite SSS measurements, and over a decade of SMOS SSS measurements. There are two primary goals of this study.

The first goal of this study is to estimate the time and length scales of SSS variability and compare the three satellite missions to check for consistency. The data from different satellite missions are often combined to create longer and more accurate SSS time series, and continuity and consistency is one of the requirements [28,29]. The second goal of this study is to provide a global map of the dominant scales of SSS variability and discuss the physical processes which might be responsible for the observed regional differences.

## 2. Materials and Methods

### 2.1. Data

2.1.1. Satellite Sea Surface Salinity

The Aquarius satellite mission, launched on 10 June 2011, included three microwave radiometers. These radiometers produced three beams at different angles. Their footprints on the sea surface, about 100 km in size, aligned across a swath about 390 km wide. Individual footprints had a sampling interval of about 10 km along the track. Aquarius's orbits had a repeat cycle of 7 days and equatorial crossings at 6 PM (ascending) and 6 AM (descending). In this study we use the Aquarius Level-2 (L2) dataset from August 2011 to June 2015, distributed by the Jet Propulsion Laboratory (JPL). Before the analysis, the L2 SSS quality status is checked following the conditions described in Melnichenko et al. [30]. In further analysis, ascending and descending tracks are processed together. The diurnal signal in SSS is very small, smaller than 0.01 PSU [31], and thus is not expected to affect the results. For a detailed description of Aquarius data, see the Aquarius User Guide.

The SMAP satellite, launched on 31 January 2015, provides near-global coverage in 3–4 days with an 8-day repeat cycle. The measuring instrument equipped with a large rotating antenna provides a different measurement approach from that of Aquarius. The SMAP radiometer provided Brightness Temperature (Tb) in a swath about 1000 km wide, with a spatial resolution of about 40 km. In this study, we use the SMAP Level-3 (L3) data from April 2015 to December 2020, provided by Remote Sensing Systems (RSS). The dataset is on a regular 0.25° × 0.25° grid created by averaging all valid L2 (swath) observations within each grid cell. The maps are produced daily by applying an 8-day running average. The effective spatial (temporal) resolution of the product is 70 km (8 days). More details on SMAP RSS V4 data can be found in Meissner et al. [32].

Launched on 2 November 2009, the SMOS satellite follows a sun-synchronous polar orbit with a 3–5-day revisit time. The instrument is a two-dimensional L-band interferometric radiometer and provides a "visible" Tb image of the ground in a swath about 1000 km wide with a spatial resolution of ~43 km. In this study, we use the SMOS L3 SSS fields from January 2010 to December 2020, generated by the Centre Aval de Traitement des Données (CATDS) Expertise Center—Ocean Salinity (CEC-OS) [33]. The debiased 9-day product temporally averaged with a half-width 9-day filter is provided every 4 days on a 25 km × 25 km grid. The effective spatial (temporal) resolution of the product is ~50 km (9 days) [33].

In the case of the Aquarius data, we take advantage of the unique Aquarius measurement geometry and analyze L2 along-track data. While gridded data may provide a convenient tool to study variability in all three dimensions [27], the mapping procedure can alter the shape of the underlying correlations and thus the inferred scales; therefore, we will use the along-track data. The results will be compared to those obtained from SMAP and SMOS satellite data which have been used in the form of gridded SSS fields. The SMAP and SMOS L2 (swath) data are much nosier and may have large gaps (particularly SMOS) [32,33]; therefore, we use L3 SSS fields which have the same grid resolution.

2.1.2. Other Data

Other observations are used to link the hydrological cycle to the dominant SSS scales. The monthly evaporation (E) data are from the Objectively Analyzed Air–sea Fluxes project (OAFlux) [34], and precipitation (P) data are from the Global Precipitation Climatology

Project (GPCP) [35]. We also use the standard Niño 3.4 index to understand the ENSO contribution to the SSS variability.

*2.2. Methods*

Following Delcroix et al. [22], the temporal and spatial scales of SSS variability are measured by the corresponding autocovariance function and defined as the e-folding time and length scales, respectively. The e-folding scale indicates a time interval or distance over which SSS variations become decorrelated and characterizes the persistence or coherence of SSS anomalies.

### 2.2.1. Estimation of the Time Scales

Time series of SSS from Aquarius observations are constructed at crossover points (Figure 1). At each crossover point, the time interval between two successive measurements (one ascending and one descending) is ~3.5 days, which technically allows resolving shorter time scales. The density of crossover points from all orbits (shown as black dots) is quite high, allowing for characterizing temporal scales of SSS variability globally with quite high spatial resolution. (The density of crossover points is slightly higher in high latitudes and lower in the tropics; the mean distance between the crossovers is ~150 km.) To suppress the instrument noise, measurements of SSS in the radius of 0.25° from each crossover point are averaged together. The resulting time series, after removing linear trends, are used to estimate the lagged autocovariances of SSS.

Examples of the SSS autocovariance function are shown in Figure 1. At each crossover point, the empirical autocovariance is approximated by a Gaussian function given in Equation (1):

$$C(\delta t) = a^2 \exp\left(-\delta t^2 / R^2\right), \tag{1}$$

where $a^2$ is the covariance at zero lag, $\delta t$ is the time lag, and $R$ is the e-folding scale. The Gaussian function has been found to best represent the lagged autocovariance in the initial segment before zero-crossing. We fit the Gaussian function to the initial segment of the empirical covariance starting from the first lag ($\delta t = 4$ days) and to the first zero-crossing. The first point ($\delta t = 0$ days), that is, the variance at zero lag $C(0)$, including both the signal and noise variance, is excluded before fitting. Our fitting procedure thus allows estimating simultaneously the unbiased signal variance, $a^2$, the e-folding scale, $R$, and the error variance, $e^2$. There, the difference between the zero-lag variance and the signal variance ($e^2 = C(0) - a^2$) is the error variance. It includes both the measurement error and the sampling error, which occur as a result of unresolved high-frequency variability [36]. The error variance and its geographic distribution are discussed in the Appendix A.

Similar approaches are applied to the SMAP and SMOS SSS data except that instead of crossover points, the time series of SSS are constructed at grid points from the gridded SSS fields. The resulting time series are used to estimate the lagged autocovariances of SSS, which, in turn, are used to estimate the dominant time scales. For consistency with the Aquarius and SMAP time series, when analyzing SMOS SSS time series of 10 years or more, a quadratic trend is removed to decrease the influence of decadal variability. Thus, our analysis of the temporal scales is restricted to scales shorter than one decade.

### 2.2.2. Estimation of the Length Scales

To assess length scales of SSS variability from Aquarius data, we consider spatial variations of SSS in a 10° by 10° (~1200 km) bin, which we move over the global ocean with steps of 5° in both the meridional and zonal directions. The size of the bin is chosen as a trade-off between the necessity to resolve the spatial scales associated with the SSS variability and the requirement of spatial homogeneity. Autocovariances are computed from the ground-track segments of the Aquarius ascending and descending passes that fit entirely into a bin (128 points with ~10 km spacing). Prior to the analysis, the multiyear mean fields (from the Aquarius L3 SSS averaged over the observation period) and linear

trends are removed from the ground-track segments. In each spatial bin, individual auto-covariances are averaged together (at least 400 in each spatial bin) to produce statistically reliable estimates. Similar to the estimation of the time scales, a Gaussian function is fitted to the empirical autocovariance to assess the dominant length scale of SSS variability, the signal variance, and the error variance. One needs to emphasize though that the geometry of the Aquarius sampling and the use of L2 data allows for estimating only the meridional length scales.

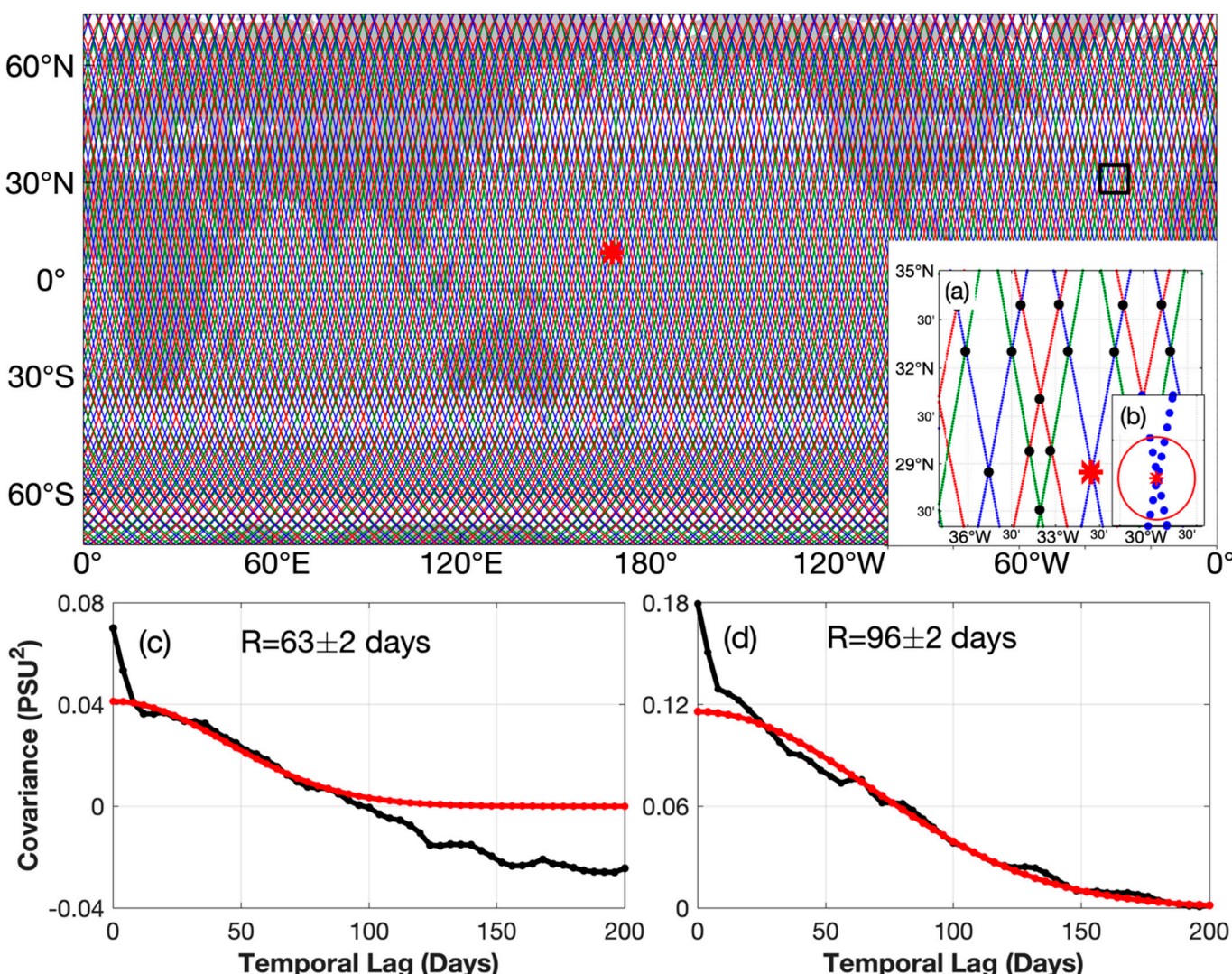

**Figure 1.** Background: Sample pattern of Aquarius ground tracks over a 7-day repeat cycle. The three beams are displayed in different colors. The combination of ascending and descending satellite passes forms crossover points shown by black dots. A zoom over the area in the subtropical North Atlantic (black rectangle) is provided in panel (**a**). Panel (**b**) shows locations of SSS measurements around one crossover point (red asterisk). To reduce the instrument noise, measurements of SSS in the radius of 0.25° from a crossover point (red circle) are averaged together. Panels (**c**,**d**) show examples of SSS autocovariance functions estimated in two regions located at the subtropical North Atlantic (left) and the tropical Pacific (right). The black curve shows the SSS autocovariance and the red curve is the Gaussian function fitted to the initial segment of the empirical autocovariance starting from the first lag (time lag = 4 days) and to the first zero-crossing.

The approach to estimate the length scales of SSS variability from SMAP and SMOS data is similar except for the use of gridded SSS fields. The spatial bins in this case are

1200 km $\times$ 1200 km in the zonal and meridional directions, which allows for estimating both the zonal and meridional scales of variability. It follows that our analysis of the length scales is restricted to scales shorter than 1200 km (scales at or longer than the measurement box cannot be resolved [37]); that is, they are in the mesoscale range and should be understood as such. Similarly, the multiyear mean SSS fields are subtracted from the weekly SSS fields to produce SSS anomalies.

## 3. Results

### 3.1. Temporal Scales of SSS Variability

#### 3.1.1. Agreement on the Global Pattern

Temporal scales of SSS variability evaluated from the three different satellite products are presented in Figure 2. The global patterns are consistent among the three satellite products, although significant quantitative differences are observed regionally. The most consistent features are observed in the relatively long SSS time scales of more than 90 days (Figure 2a,c,i, thin contours). The longest time scales, up to 160 days (bold contours), indicate that the variability in these regions is dominated by the inter-annual signal. They are observed in the ENSO-related regions of the central and western tropical Pacific (WTP) and along the SPCZ [38–45]. The shortest scales of SSS variability, shorter than 60 days, are likely controlled by fast-moving features such as eddies. They are found mostly in the Southern Ocean between approximately 30°S and 50°S, the North Indian Ocean, and in some regions of the western subtropical Pacific and Atlantic (Figure 2a,c,i). Remarkably, the eastern part of the equatorial Pacific and equatorial Atlantic have relatively short time scales, emphasizing the importance of short-term variability in the region, likely associated with Tropical Instability Waves (TIW, [24,46]).

Overall, the global distribution of SSS temporal scales spans from ~15 days in the Southern Ocean to more than 90 days in the tropics, generally consistent with the analysis of the first three years of Aquarius observations by Bingham and Lee [27]. Our results also highlight that one prominent area of enhanced inter-annual variability lies in the Northeastern Pacific. It occurs mostly in SMAP observations (Figure 2c) and to a lesser extent in SMOS observations over the same period (Figure 2g).

Earlier studies have emphasized the importance of the seasonal cycle in SSS variability [48]. In regions where the seasonal cycle is not dominant, other variabilities may take the lead, either short-term variability or inter-annual variability. We remove the seasonal cycle to confirm the variation in temporal scales associated with the dominant variability. The seasonal cycle explains more than 25% of the SSS variance and plays an important role in the tropical belt (Figure 2, area marked with black dots). The seasonal cycle's removal from the time series has two interesting consequences. In the areas dominated by the inter-annual variability (scales longer than 90 days), once the seasonal cycle is removed, the temporal scales become longer, such as in the WTP, over the SPCZ, and in the Northeastern Pacific (compare left and right panels of Figure 2). On the contrary, areas with relatively short temporal scales (<60 days) expand, while their temporal scales decrease. This effect was also observed in previous studies using Aquarius data (e.g., [27]) and is widely found in the three satellite products. However, the very short time scales (~15 days) observed in some parts of the ocean when the seasonal cycle is removed, particularly in the Southern Ocean, may indicate a high level of uncorrelated noise and may also include unresolved temporal variability (as discussed in the Appendix A).

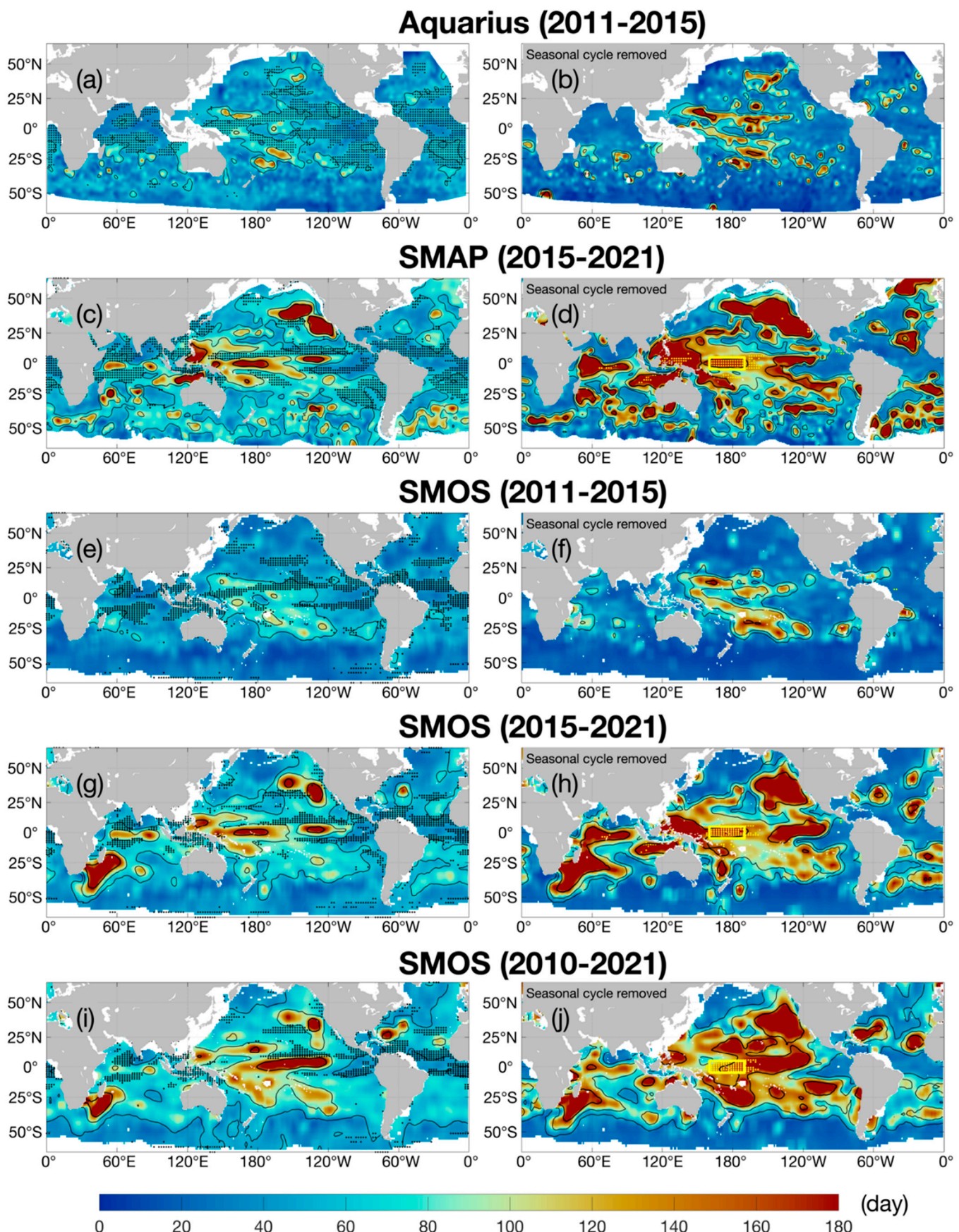

**Figure 2.** Temporal scales (days) of SSS variability derived from (**a**,**b**) Aquarius, (**c**,**d**) SMAP, and (**e**–**j**) SMOS satellite data. The SMOS dataset is split into three segments: (**e**,**f**) covering the Aquarius

observation period (2011–2015), (**g**,**h**) SMAP observation period (2015–2021), and (**i**,**j**) the whole SMOS observation period from 2010 to 2021. Left (right) panels show the scales estimated before (after) the annual cycle is removed from the SSS time series. Contours are 60, 90, and 150 days. Black dots mark regions where the annual cycle contributes more than 25% to the total SSS variance. Yellow dots mark regions where ENSO-related SSS variability contributes more than 15% to the raw SSS variance. We define the ENSO-related part as the SSS variability regressed to the Niño 3.4 index. The key region in Figure 2d,h,j is defined as (5°S–5°N, 160°E–170°W). Gaps in coverage, particularly in the Aquarius data in the eastern North Atlantic, are due to the data corrupted by radio frequency interference (RFI) contamination [47].

### 3.1.2. Regional Differences and Possible Reasons

It is interesting to note that despite visual similarity in the spatial patterns of the SSS temporal scales evaluated from Aquarius and SMAP data (Figure 2a–d), there are quite large quantitative differences, particularly in the Northeastern Pacific, where we can observe long time scales in SMAP data but not in Aquarius data. In many places the SMAP scales are longer than 150 days, notably in the WTP, over the SPCZ, in the northeast and southeast Pacific, indicating strong inter-annual variability.

The differences between the Aquarius and SMAP estimates can be related to two factors. First, they cover different observation periods while the processes governing SSS variability are non-stationary. Second, there are differences in data sampling, temporal resolution and/or degree of smoothing. As the SMOS satellite mission provides a continuous data record from 2010 to present, covering both the Aquarius and SMAP observation periods, it is possible to verify the estimates with a quasi-independent data source. For this, we split SMOS data into two segments covering, respectively, the Aquarius and SMAP observation periods, and repeat the analysis. The results are shown in Figure 2e–h. The estimates achieve a remarkable degree of agreement between the Aquarius and SMOS over the period 2011–2015, as well as between SMAP and SMOS over the period 2015–2021, respectively, indicating that the processes governing SSS variability are indeed non-stationary and a transition to a stronger inter-annual variability occurred during 2015 (compare Figure 2a,b to Figure 2e,f; Figure 2c,d to Figure 2g,h). This is likely related to the 2015/2016 super El Niño event (the warm phase of ENSO) which reached remarkable magnitudes [49,50]. It was captured by SMAP and SMOS; however, it was missed in Aquarius because the latter ended operations in June 2015. This strong inter-annual signal in the tropics presumably dominated the temporal scales of SSS variability during the SMAP observation period (2015–2021). Computed over the whole duration of SMOS observations from 2010 to 2021, the area of long temporal scales in the tropics expands greatly (Figure 2i,j), indicating the significance of the inter-annual and longer-period variability. This dominance of the low-frequency signal is consistent with the redness of the SSS spectrum [51].

One key region of the tropical Pacific is of particular interest since it achieves excellent agreement on time scales of SSS variability in all three satellite datasets. This area, located within the Niño 4 region in the central tropical Pacific, has the strongest signal in the inter-annual SSS variability, and has been connected to the zonal shift of the SSS front in ENSO evolution (e.g., [52]). Our estimates show that the ENSO-related SSS variability can contribute more than 15% to the total SSS variance in that area, although some discrepancy exists between the two satellites (Figure 2d,h,j; yellow dots mark regions). Next, we specifically consider the ENSO-affected SSS persistence there (see Figure 2d,h,j, yellow box).

We focus on the 2015/2016 super El Niño event and attempt to quantify the persistence of SSS during its two-year evolution (ENSO-involved year, Figure 3, red lines). For comparison, time scales in years without ENSO events (called the non-ENSO-involved year) are also estimated using the other two-year SSS anomaly segments from 2010 to 2021 (e.g., 2013–2014, Figure 3, gray lines). Here we only use SMOS data due to their long observation period. Figure 3a shows SSS covariance functions in different situations. In the

ENSO-involved year we observe longer SSS persistence in the order of 90 days. While in the non-ENSO-involved year, SSS persistence falls below the temporal scale of about 43 days (Figure 3a, black line). Removing the seasonal cycle has a generally similar effect on the persistence of SSS anomalies (Figure 3b). These results are consistent with our expectation that persistence influenced by the ENSO causes differences in the temporal scales observed by satellites (e.g., Aquarius and SMAP).

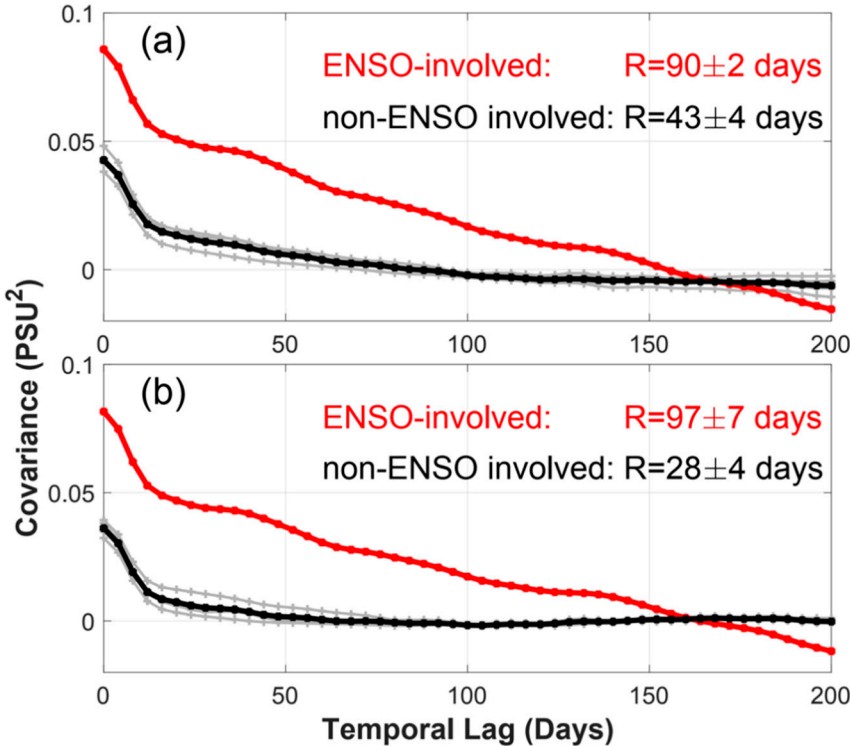

**Figure 3.** (**a**) Temporal autocovariance functions of SSS variability in the Niño 4 region. The red curve corresponds to the SSS time series from January 2015 to December 2016. The gray curves indicate the time series over 2011–2012, 2013–2014, 2017–2018, 2019–2020, and the black curve represents their ensemble-mean. As mentioned in Section 2.2.1, the temporal scale is defined as the e-folding decorrelation scale of the Gaussian function (not shown) fitted to the autocovariances. The region where the autocovariances are computed is indicated by the yellow rectangle in Figure 2. (**b**) as in (**a**) but with the annual cycle removed.

### 3.2. Spatial Scales of SSS Variability

3.2.1. Agreement on the Global Pattern

The spatial scales of SSS variability show how the SSS fluctuations are coherent as a function of distance. The meridional scales are obtained from the three datasets, while the zonal scales are available only from SMAP and SMOS data. They are displayed in Figure 4. Qualitatively, the distributions of the zonal and meridional scales are very similar in the three satellite datasets. In the three datasets, the meridional length scales of SSS variability are generally longer (up to 150 km) in the tropics and shorter (<60 km) in the subpolar regions (Figure 4a,c,e). However, the global distribution of the spatial scales is not simply latitudinal but exhibits a more complex spatial pattern. In the topical Pacific, the quasi-zonal band of relatively long meridional scales stretches from the eastern basin to the western fresh pool following the mean position of the ITCZ (along ~3°N–5°N, thin dash contours), emphasizing the role of external forces in setting the scales. This pattern is particularly visible in the SMOS data, while Aquarius and SMAP show a secondary maximum south of the Equator along ~5°S. Two findings should be noted. One of the interesting results is that the meridional scales tend to be a little longer (by about 20%) in

the eastern tropical Pacific (ETP) but relatively shorter in the western basin, contrary to the distribution of temporal scales. This indicates that the ETP has both short-term SSS persistence (<60 days) and long-distance SSS coherency (up to 150 km) occurring in the same region. These features are possibly characterized by the TIW-induced SSS anomalies (high-frequency) and the ENSO-induced SSS signature (large-scale). Another interesting result is that we observe relatively long meridional scales in the SPCZ region, characterized by a large-scale SSS front and strong inter-annual variability [27,52,53].

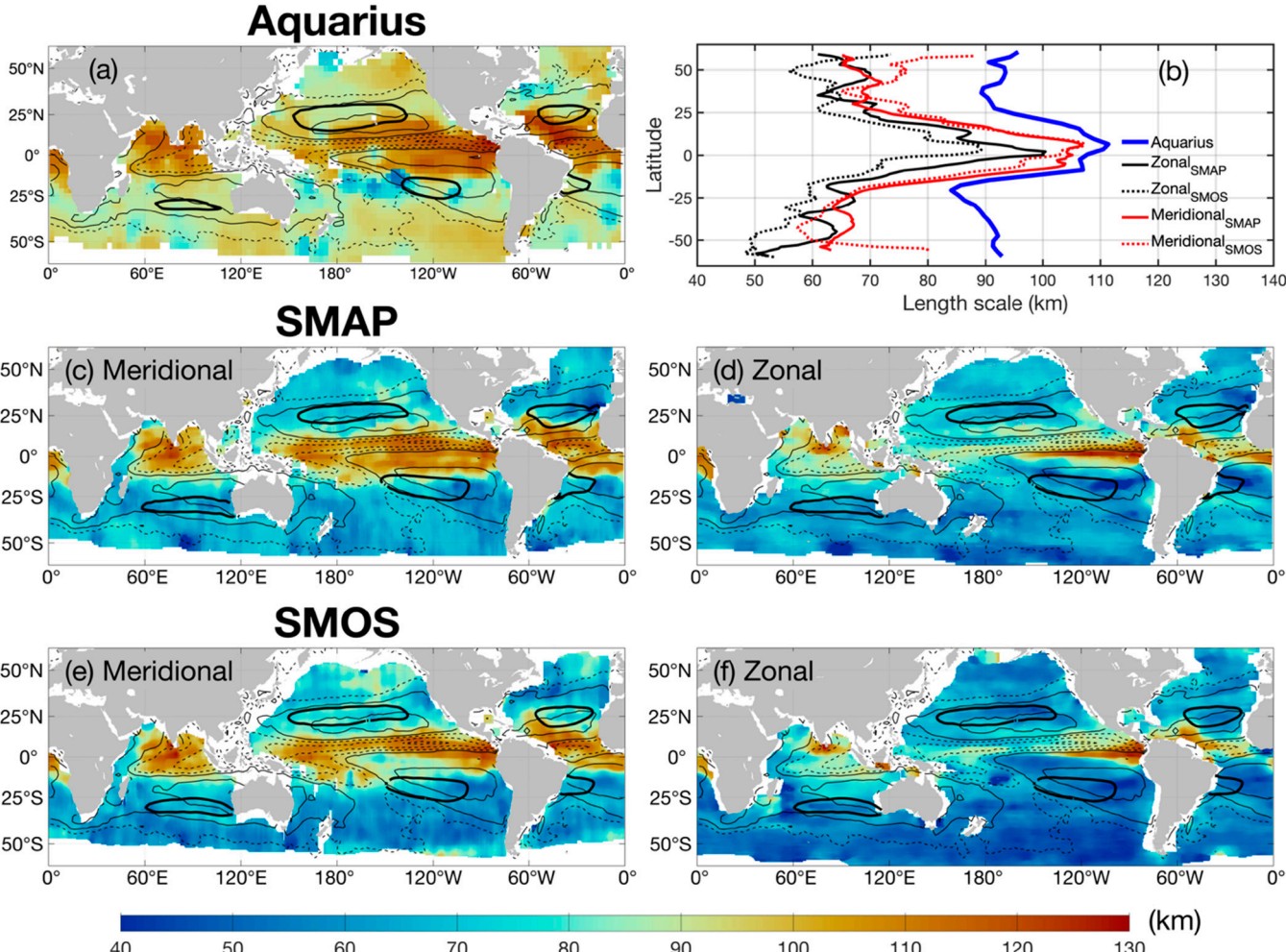

**Figure 4.** Spatial scales of SSS variability derived from (**a**) Aquarius, (**c**,**d**) SMAP, and (**e**,**f**) SMOS satellite missions. For the latter two, spatial scales are calculated in the zonal and meridional directions. Bold contours circle local SSS maxima with isohalines in the North Pacific (35.2 PSU), South Pacific (36.0 PSU), South Indian Ocean (35.6 PSU) and Atlantic Ocean (37 PSU). Thin contours in (**a**) show the climatological mean evaporation minus precipitation $(E - P)$. Solid (dashed) contours indicate positive (negative) $E - P$; zero contour is omitted. C. I. is 2.5 mm day$^{-1}$. Zonally averaged length scales are displayed in (**b**).

Similar behavior is observed for the zonal scales (Figure 4d,f). Relatively long zonal scales (~150 km) are observed in the ETP, in contrast to the WTP, where the zonal scales are much shorter (90–100 km). Relatively long zonal scales are also observed in the tropical Atlantic, particularly in its western part, and in the eastern and northern tropical Indian Ocean. In the interiors of the subtropical gyres (bold contours) the spatial scales are relatively short and generally isotropic; there is no significant difference between the zonal and meridional scales.

The distribution of the spatial scales is slightly asymmetric about the equator with the longest spatial scales observed in the Northern Hemisphere around 5°N–10°N. The meridional variations in the distribution of the spatial scales are clearly apparent in the zonal averages shown in Figure 4b. The zonal averages also illustrate the excellent agreement between SMOS and SMAP in the estimates of the meridional scales (the estimates start to diverge in high latitudes, poleward of about 50°), and quantitative differences in the estimates of the zonal scales. The zonal scales estimated from the SMAP data are a little longer (by about 20%) than those estimated from the SMOS data. The possible reason for this discrepancy is not clear but can be related to the differences in the design and sampling strategy between the two satellites.

Our analysis of the SMAP and SMOS satellite data also reveals that the spatial scales of SSS variability are highly anisotropic. The meridional scales in the tropical belt are typically longer than the zonal ones and become more isotropic towards higher latitudes (Figure 4b). The exceptions are in the ETP and western tropical Atlantic (WTA), where the zonal scales are slightly longer (by about 20%) than the meridional ones (Figure 4d,f). Given the predominantly zonal distribution of the large-scale SSS, the zonal scales of SSS variability reflect presumably the eddy length scales, which drop monotonically from the equator toward the high latitudes [54]. This would be generally consistent with our estimates of the SSS zonal scales in Figure 4b. Likewise, a broad similarity exists between the estimates from the three satellite missions, showing the anisotropic pattern of the SSS spatial scales.

Removing the seasonal cycle has generally little effect on the estimated spatial scales (Figure 5). The largest differences are in the tropical belt where the annual cycle is the dominant component of SSS variability. There, the variability becomes more isotropic when the annual cycle is not included although this effect is observed primarily in SMAP data.

### 3.2.2. Regional Differences and Possible Reasons

Although the spatial distributions of the dominant length scales estimated from the three data sets are very similar qualitatively, there are noticeable quantitative differences. The meridional scales estimated from Aquarius data tend to be longer in the tropical region compared to those estimated from SMAP and SMOS (Figure 4a vs. Figure 4c,e). At the same time, Aquarius measurements tend to overestimate the meridional scales in the subpolar regions. The reason for these discrepancies is likely because of the different spatial resolution of the satellite datasets and/or the different level of noise. For example, the Aquarius footprint is ~100 km compared to ~40 km of SMAP and SMOS. This relation is not obvious or straightforward, however, as the meridional scales estimated from SMAP and SMOS data demonstrate remarkable similarity while the zonal scales are systematically smaller in SMOS data. The differences in design and sampling strategy between the different satellite platforms may also play a role.

One also needs to emphasize that the effective spatial resolution (feature resolution) of the gridded SMAP and SMOS SSS fields used to estimate the spatial scales is around 60 km. Therefore, scales shorter than 60 km estimated from the maps in some areas indicate the dominance of noise or, in other words, a large noise-to-signal ratio. Such areas are observed mostly in the subpolar regions, particularly in the Southern Ocean poleward of ~50°S, where errors in SSS maps are indeed large (Figure A1).

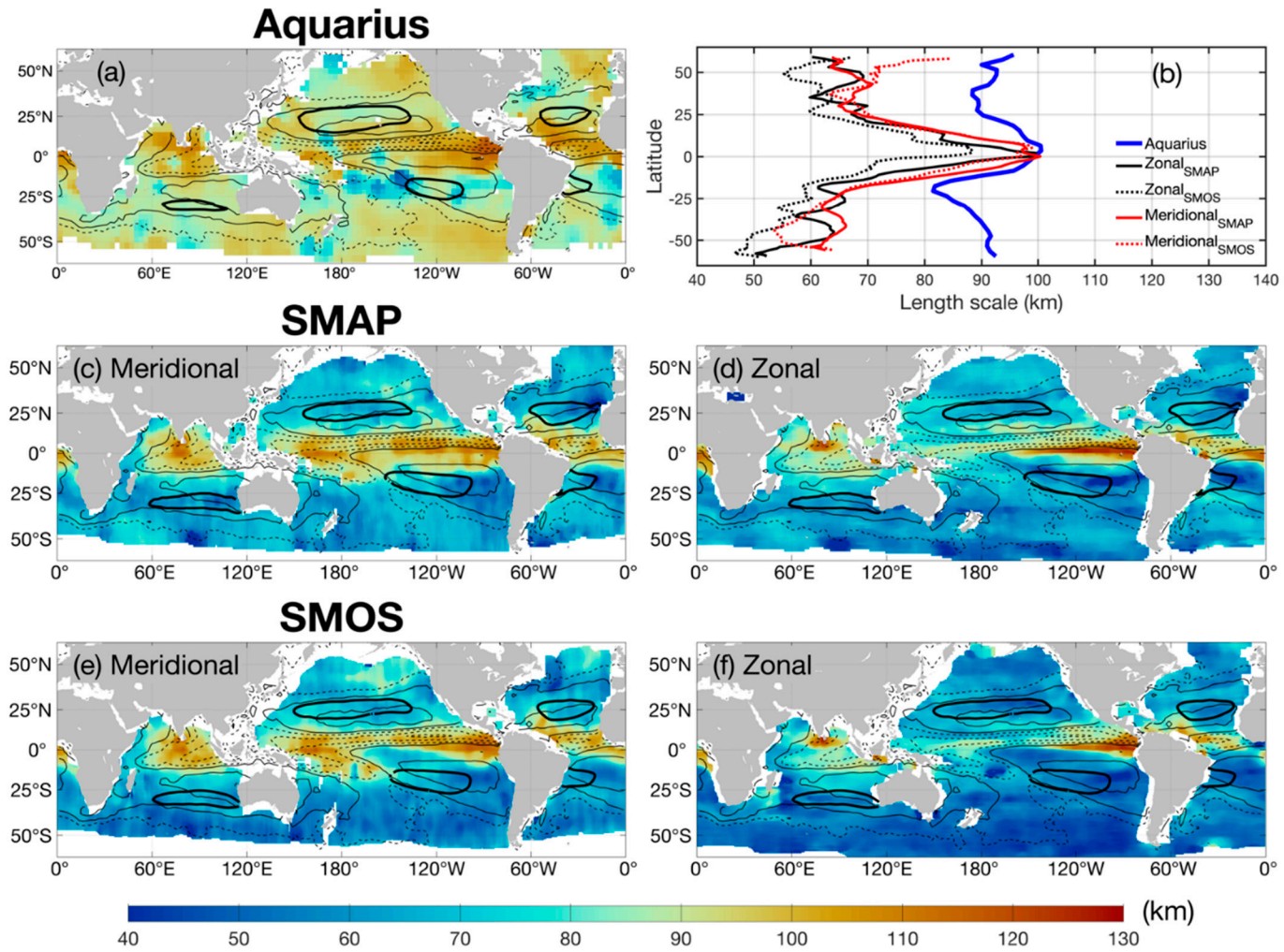

**Figure 5.** Spatial scales of SSS variability without the annual cycle, as in Figure 4.

## 4. Discussion

This study aims to evaluate and compare the time and length scales of global SSS variability using satellite SSS observations, including SMOS, Aquarius, and SMAP satellite missions. These characteristic scales are defined as the e-folding time and length scale, respectively, of the corresponding autocovariance functions. Overall, there is general agreement between the three datasets showing the same global patterns of the time and length scales of SSS variability, although noticeable quantitative differences exist.

The dominant temporal scales of SSS variability vary between more than 90 days in the tropics and ~15 days in the high latitudes (e.g., in the Southern Ocean). The very short temporal scales (close to the Nyquist period) in some parts of the ocean are probably due to the high level of noise in the satellite data or the high noise-to-signal ratio. The longest temporal scales (up to 160 days) are observed in the central and western tropical Pacific, as well as along the SPCZ. These areas are also related to the strongest ENSO-related signal in SSS and achieve good agreement in the three satellite missions. These SSS anomalies sustain from season to season and may be useful for regional climate predictions, e.g., as a precursor for predicting rainfall in the Australian and East Asian regions [55,56].

The results of nonseasonal SSS variability support the dominance of the inter-annual variability in these regions. With the removal of the seasonal cycle, time scales of more than 90 days become prominent over the tropical belt up to 30° of latitude. The relative importance of decadal- and longer-period variability is not addressed in this study because the satellite data records are not long enough to resolve these signals. In this regard,

the estimated temporal scales are thought to characterize the so-called short-term SSS persistence [57]. We can also see that the processes governing the SSS distribution and variability are non-stationary, as evidenced by the differences in the dominant temporal scales of SSS variability estimated over different periods, 2011–2015 and 2015–2021, ENSO-involved and non-ENSO involved years. Overall, the global distribution of the SSS temporal scales is consistent between the three satellite SSS datasets and in agreement with that inferred from the first three years of Aquarius observations by Bingham and Lee [27]. Our analysis of SMAP and SMOS observations also highlights another prominent area of enhanced inter-annual variability lying in the Northeastern Pacific. The available data are somewhat limited and no previous study has focused on this large area of long temporal scales. However, the understanding of inter-annual and/or longer-timescale SSS variability in mid-latitudes is not complete and needs further assessment.

The geographical distribution of the spatial scales of SSS variability estimated from the three satellite SSS datasets is very similar qualitatively, although there are noticeable quantitative differences. The dominant spatial scales of SSS variability are generally the longest (up to 150 km) in the tropics and the shortest (<60 km) in the subpolar regions. However, the distribution of the dominant spatial scales is not simply latitudinal but exhibits a more complex spatial pattern. In the tropics, there is slight east-west and inter-hemispheric asymmetry in the spatial scales, which is observed in the Pacific but is absent in the other two oceans. In particular, the spatial scales tend to be a little longer, by about 20%, in the ETP but relatively shorter in the WTP, contrary to the distribution of the temporal scales. In addition, relatively long spatial scales (up to 120 km in the meridional direction) are observed along the SPCZ, where a large-scale SSS front and strong inter-annual signals in SSS are also observed (e.g., [27]).

Our analysis of SMOS and SMAP SSS data reveals that the spatial scales of SSS variability are highly anisotropic. The zonal components are typically shorter than the meridional ones in the tropical belt (except for the ETP and WTA) and become more isotropic towards higher latitudes. This is in contrast to earlier studies (e.g., [38,58,59]) which describe the zonal scales as being generally longer than the meridional ones, particularly in the tropics. The explanation is in the definition of the spatial scales (see Section 2.2.2). First, our analysis of the SSS spatial scales is, by design, restricted to scales shorter than 1200 km (the size of the moving window); that is, they are largely in the mesoscale range. Second, the spatial scales here are estimated as the spatial autocovariance (using instant SSS maps) and not as correlations between the time series at two distant locations (e.g., [58]). The two approaches are not the same (think of it as a wavenumber and wavenumber-frequency spectra), which may explain the discrepancy. Yet, this kind of information on the spatial scales is what is typically needed in many practical applications, particularly for objective interpolation of irregularly spaced satellite and in situ measurements and for blending observations from different satellite platforms (e.g., Aquarius and SMOS).

Regional differences in the estimates of the scales from the three satellite SSS datasets may arise as a result of discrepancies in the spatial resolution and/or different noise levels. Perhaps just as important, the satellite data are subject to biases, including large-scale and time-varying biases [60], which may affect the estimates of the scales. The duration of time series is another limitation in the scale estimates (as discussed in Section 3.1).

There are many unanswered questions about how the interplay between the various ocean processes controlling the SSS distribution and variability result in the observed spatial and temporal scales. This might be an important issue for future research.

## 5. Conclusions

Time and length scales of SSS variability are characterized globally from the data of three satellite SSS missions, SMOS, Aquarius and SMAP. Our results show that:

- The geographic patterns of the time and length scales of SSS variability are generally consistent between the three satellite missions, although there are noticeable quantitative differences. The differences are likely due to differences in the design and

sampling strategy between the satellite missions and/or different level of noise in the data.

- The temporal scales of SSS variability vary from more than 90 days. The longest time scales (up to 160 days) are observed in the western tropical Pacific and are related to the ENSO variability. The very short time scales (close to the Nyquist period) in some parts of the ocean are likely due to high levels of noise in the data (high noise-to-signal ratio).

- The longest-length scales are in the tropics (with slight asymmetry around the Equator such as the longest scales are observed in the North Hemisphere around 5°N–10°N) and decrease towards higher latitudes.

- The length scales are anisotropic in the tropics (the zonal scales are generally shorter than the meridional ones) and become isotropic towards higher latitudes.

- The processes governing the SSS distribution and variability are non-stationary. The complex spatial patterns of the time and length scales of SSS variability seem to reflect the underlying physical process governing the variability.

**Author Contributions:** Conceptualization, O.M. and D.L.Y.; methodology, O.M.; analysis, D.L.Y.; writing—original draft preparation, D.L.Y.; writing—review and editing, D.L.Y., O.M., P.H. and K.F.; funding acquisition, O.M. and K.F. All authors have read and agreed to the published version of the manuscript.

**Funding:** This research was funded by the National Key Research and Development Program of China (Grant No. 2022YFE0106800), the Natural Science Foundation of China (NSFC) (No. 41730964 ), the Research Council of Norway funded project (MAPARC, No. 328943), the Innovation Group Project of the Southern Marine Science and Engineering Guangdong Laboratory (Zhuhai) (No. 311020001), the National Aeronautics and Space Administration (NASA) Physical Oceanography Program (Nos. NNX17AK06G and 80NSSC20K1338), the Fundamental Research Funds for the Central Universities, Sun Yat-sen University (No. 74110-31610065).

**Data Availability Statement:** The Debiased V5 SMOS SSS product is freely distributed at https: //www.catds.fr/Products/Available-products-from-CEC-OS/CEC-Locean-L3-Debiased-v5. The JPL version 5.0 Aquarius SSS product is available for public access at https://podaac-tools.jpl.nasa. gov/drive/files/allData/aquarius/L2/V5/SCI. The Aquarius L3 SSS used for mapping is available for public access at https://podaac-tools.jpl.nasa.gov/drive/files/allData/aquarius/L3/mapped/ V5/7day_running/SCI. The RSS V4 SMAP SSS data can be downloaded at http://www.remss. com/missions/smap/. The Niño 3.4 index is obtained from ftp.cpc.ncep.noaa.gov/wd52dg/data/ indices. Open access monthly surface freshwater flux data sets: Evaporation—OAflux (http://apdrc. soest.hawaii.edu/datadoc/whoi_oafluxmon.php), Rainfall—GPCP (http://apdrc.soest.hawaii.edu/ datadoc/gpcp_monthly.php). All data were accessed 18 October 2022.

**Acknowledgments:** The research was conducted while Daling Li Yi was a Postdoctoral Fellow at the International Pacific Research Center (IPRC) of the University of Hawaii. IPRC/SOEST contribution 1582/11591. The authors would like to thank three anonymous reviewers for their useful comments.

**Conflicts of Interest:** The authors declare no conflict of interest.

**Appendix A Error Variance of Satellite SSS Dataset**

The error variance, $e^2$, is calculated from empirical covariances of the SSS anomalies as the difference of the data variance at zero lag, $C(0)$, and the signal variance, $a^2$, estimated by the y-intercept of the functional fit to the empirical covariance: $e^2 = C(0) - a^2$. Two examples are presented in Figure 1. The error variance includes both the measurement error (assumed to be random) and the sampling error, which occurs as a result of unresolved high-frequency variability [36].

Maps of the estimated root mean square (r.m.s.) of the uncorrelated error in the SSS time series for the three satellite datasets are presented in Figure A1. All three satellites show consistent patterns. The r.m.s. error is typically smaller in the tropical regions (warm water) and larger, up to 0.5 PSU, in high latitudes (cold water). Larger errors are also observed in the coastal areas, particularly along the Eurasian continent, possibly owing to

radio frequency interference (RFI) contamination [47,60]. Overall, in the mid-latitudes and the tropics (40°S–40°N), the Aquarius satellite performed better (Figure A1a), followed by SMAP (Figure A1b) and SMOS (Figure A1c). In high-latitude oceans, particularly near the Antarctic, the time series from SMOS showed considerably smaller r.m.s. errors, compared to Aquarius and SMAP. This result is a bit surprising but can be related to additional spatial/temporal smoothing in SMOS SSS maps. Another prominent feature observed in all three error maps is a quasi-zonal band of high error variance coinciding with the mean locations of the North Atlantic, Pacific ITCZ and the SPCZ. These bands of elevated error variance, given their locations, are likely due to unresolved high-frequency variability rather than instrument error. Taking the surrounding areas as the background, we can estimate that the r.m.s. of unresolved high-frequency variability (periods shorter than ~15 days) is around 0.1 PSU.

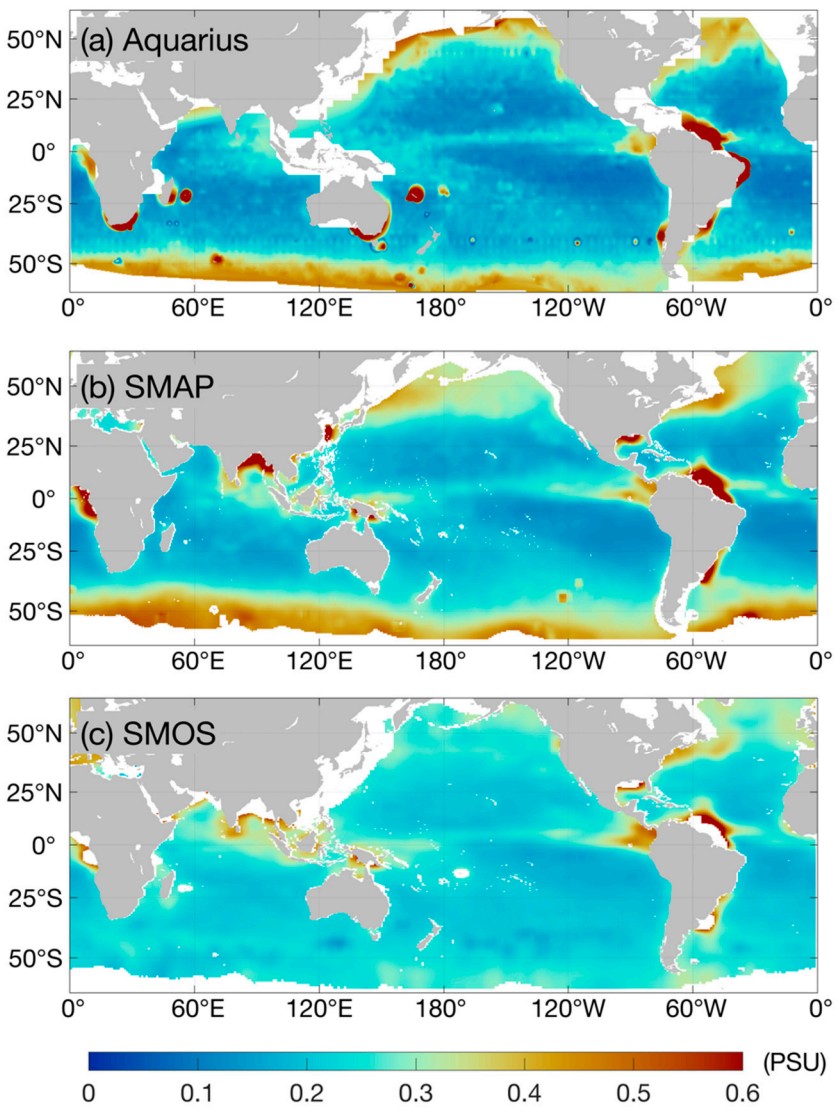

**Figure A1.** Root mean square of the uncorrelated error estimated from (**a**) Aquarius, (**b**) SMAP and (**c**) SMOS SSS data.

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
