# Peer review of "Satellite-Observed Time and Length Scales of Global Sea Surface Salinity Variability: A Comparison of Three Satellite Missions"

_remotesensing, doi:10.3390/rs14215435_

Round 1
Reviewer 1 Report
Review of “Satellite-observed time and length scales of global sea surface salinity variability: a comparison of three satellite missions”
Overall, the paper is scientifically sound and logically organized. This paper provides an interesting comparison of scales of sea surface salinity (SSS) variability as seen by the past (Aquarius) and present (SMAP and SMOS) missions. The comparison is done carefully, clearly and comprehensively, the results extend previous studies and address these scales are important to quantify, as they are used in optimal interpolation methods to build gridded and continuous SSS products.
On one aspect, an agreement on the three satellite missions makes it a useful reference for the statistic and dynamical analysis of the SSS variability from a global perspective. On another aspect, this paper provides reasonable explanations for differences caused by data, methods, definition, nonstationary signals and etc.. There, I would recommend the paper for publication after addressing the following minor comments.
Comments:
1, In the method section, why authors use Gaussian function to fit the SSS autocovariance function? Why don’t you choose other function? Please explain it using one sentence.
2, In Figure 2, why does Aquarius map has some gap in the subpolar region in the North Atlantic (Figure 2a, b)? Is there any data quality control? And what’s the main reason for that? I’d suggest to add a little bit description.
3, In Figure 4 and 5, in my eyes, the spatial pattern of Aquarius results are quite different with that of two satellites, why don’t use Aquarius L3 data rather than L2 along-track data for consistency?
4, L175, “…in the radius of 0.25o from..”, the unit of degree should be revised. Same problems are found in L230, L248.
5, L246, as mentioned in the paper, the interannual SSS variability is related to the ENSO-related studies. Although historical observation is limited, longer outputs from numerical model are used in the studies of ENSO-related SSS variability recent years. Please add some model reference in those ENSO-related regions.
6, L289, figure legend, “NINO 3.4 index”, should be revise to “Niño 3.4 index”, try to be consistent in the paper.
7, L522, “An example is presented in Figure 1.” Revise it to “Two examples are …”
8, L532, “significantly smaller …” try to avoid the term if it is not associated with a statistical test.
Reviewer 2 Report
The article consists of 4 parts, each of which is logically structured and sequentially reveals each stage of research and comparison of data. The introduction contains both information about the degree of knowledge of determining the scale of SSS variability in the World, current and information related to research issues at the present time.
The Materials and Methods section provides a concise description of the 3 Aquarius satellites, SMAP and SMOS, with cycle times of 7, 8 and 9 days.
The figures are of high quality and fully reveal all stages of the study.
I noticed only 1 small nuance (typo): Line 39 (in the reference to the literature, correct the second 1. Should be [1,2]
To crown it all: the article certainly deserves publication in this thematic collection and will be of interest to both narrow specialists and those who makes complex researches, including the analysis of natural and satellite data.
Author Response
Response to Reviewer #2 of our manuscript, entitled “Satellite-observed time and length scales of global sea surface salinity variability: a comparison of three satellite missions”, by Daling Li Yi et al.
1, I noticed only 1 small nuance (typo): Line 39 (in the reference to the literature, correct the second 1. Should be [1,2].
We revised it, see Line 37, thanks.
Reviewer 3 Report
The manuscript entitled “Satellite-observed time and length scales of global sea surface salinity variability: a comparison of three satellite missions” by Yi et al. represents a significant contribution to the SSS variability and its temporal and spatial distribution in the global oceans. As indicated by the authors, the global coverage of this work with characteristic examples through the comparison of 3 satellite missions for each described ocean makes this manuscript potentially important. I found the topic of the manuscript very interesting and possibly of great interest for the Remote sensing readers.
Overall, all data are sufficient, and the treatment of the data are appropriated. The figures are appropriated as both quantity and quality. The length of this review paper is appropriated for this journal, with all interpretations and conclusions to be in general very well justified. The text is very well organized, and this makes the manuscript easily readable and understandable. The bibliography is accurate and updated too. The English is in relatively good shape. However, there are some minor points that need to be clarified and/or better discussed before acceptance. Therefore, I propose some minor points to be addressed before it can be considered for publication (minor revision). Both the minor comments and suggestions are listed right below.
My main concern is about the global coverage of this approach. The Mediterranean Sea is not included in this comparison even though this oceanic basin presents strong salinity variability from west to east. Did the authors have any data for the SSS distribution pattern within this basin? The incorporation of such data will enhance their findings in a more global way. I would suggest trying to assess these data if are available.
L109-111: The last paragraph is not needed here, since it is a scientific article and not a book chapter where this description would fit more than here. I suggest deleting it.
I would also suggest separating the discussion from the conclusions
